# Maximum Entropy-Minimum Residual Model: An Optimum Solution to Comprehensive Evaluation and Multiple Attribute Decision Making

**DOI:** 10.3390/e27020203

**Published:** 2025-02-14

**Authors:** Qi-Yi Tang, Yu-Xuan Lin

**Affiliations:** 1Institute of Insect Sciences, Zhejiang University, Hangzhou 310028, China; 2Faculty of Science and Technology, BNU-HKBU United International College, Zhuhai 519087, China; yuxuanlin@uic.edu.cn; 3Guangdong Provincial Key Laboratory of Interdisciplinary Research and Application for Data Science, BNU-HKBU United International College, Zhuhai 519087, China

**Keywords:** composite indicator, comprehensive evaluation, entropy, multiple attribute decision making

## Abstract

To assess a subject with multiple factors or attributes, a comprehensive evaluation index, or say a composite indicator, is often constructed to make a holistic judgement. The key problem is to assign weights to the factors. There are various weighting methods in the literature, but a gold standard is lacking. Some weighting methods may lead to a trivial weight assignment that is one factor having a weight equal to 1 and the others all zero, while some methods generate a solution contradicting intuitive judgement, or even infeasible to calculate. This paper proposes a new model to generate weights based on the maximum entropy-minimum residual (MEMR) principle, directly estimating the relationship between factor weights and the composite indicator. The MEMR composite indicator extracts the common feature of multiple factors while preserving their diversity. This paper compares the MEMR model with other commonly used weighting methods in various case studies. The MEMR model has more robust, consistent, and interpretable results than others and is suitable for all comprehensive evaluation cases involving quantitative factors. The optimization technique of the proposed MEMR model and the related statistical tests are included as a package in the DPS (data processing system) software V21.05 for the convenience of application in all fields.

## 1. Introduction

Comprehensive evaluation is to assess a subject based on multiple factors or dimensions. It often collaborates with the methods of multiple criteria decision making (MCDM) or multiple attribute decision making (MADM), quantifying and assigning weights to each criterion (or attribute) in order to determine the best decision or choice, providing a holistic judgment. In complex decision-making environments, it facilitates making the optimal decision. Comprehensive evaluation is widely used in emerging strategic fields such as the humanities, environmental protection, technology, diversity, and sustainable development. In the process of constructing the comprehensive evaluation index (CEI), how to reasonably assign weights to the corresponding factors is a key consideration. The weight of each evaluation factor indicates its role and reflects the value orientation of the evaluation subject; it is the core to work out scientific, reasonable, and fair results. Therefore, the weighting method is a key consideration in research on multi-factor comprehensive evaluation. In this regard, ref. [1] systematically reviewed the problems associated with weighting, aggregation, and robustness in CEI construction. In the existing weight assignment methods for CEI, there is no recognized standard. For the same evaluation subject, adopting various comprehensive evaluation methods leads to different ranks and judgments. There is no existing perfect weighting method since each approach has its advantages and limitations. It is an inevitable problem in the field of CEI to be solved.

The handbook published by OECD/EU-JRC in 2008 [2] introduces various weighting methods for CEI calculation, outlining the advantages and disadvantages of each. The lack of a unified weighting method has led some researchers [3,4] to question the rationality and fairness of multi-factor comprehensive evaluation, suggesting that subjectivity is a central defect. In the handbook [2], the authors dedicate a considerable amount of space to the benefit of doubt (BoD) method proposed by [5,6,7] with reference to the data envelopment analysis (DEA) model as a tool for handling weights in comprehensive evaluation analysis. However, BoD has some serious theoretical problems. For example, some components might be 0 in the weight calculation results, and the extreme situation in which the weight of one factor is not 0 while the weight of others is 0 might arise (i.e., the 0–1 weight problem). The reason for this problem is the loose weight constraints in the original BoD model, resulting in some weight components becoming 0, while others have excessively large weight loads. Since BoD is a weight calculation model that evaluates the relative advantages of evaluation objects through rewards, the weight calculation results can flexibly change based on the information from the evaluation object set, ensuring that each evaluation object receives the optimal score calculation result. However, when a certain factor of an evaluation object stands out too much compared to other evaluation objects, the weight share of that factor may be much higher than other factors, leading to some factors having zero weight. In extreme cases, it may result in a situation where all weights are 0 except for the most salient factor, which is an overly flexible weight phenomenon, also referred to as the overly soft weight phenomenon or the 0–1 weight phenomenon. Specifically, when the number of evaluation objects and factors are close, the frequency of this extreme phenomenon increases. Moreover, BoD models have multiple solutions. In addition, since the weight vectors of each evaluation object in the original BoD model are different, and the sum might not be equal to 1, CEI calculated by BoD is theoretically not comparable.

Especially in China, with the development of the market economy, the existing comprehensive evaluation methods are becoming increasingly inadequate in providing decision-makers with accurate measurements to assess the real situation of economic and social development in various regions under the new circumstances. At the same time, for those evaluation subjects affected by low rankings, if we do not have an objective and standardized comprehensive evaluation system, the evaluated subjects may have reasons to question the fairness and rationality of the evaluation results, which could lead to a reduction in the authority of the comprehensive evaluation. With the rapid growth of available information, CEIs are becoming more widely used. For example, ref. [8] determined more than 400 CEIs for ranking or evaluating countries based on economic, political, social, and environmental factors. Meanwhile, ref. [9] recorded more than 100 CEIs of human progress in a report by the United Nations Development Program. Although these lists are still far from being exhaustive compared to the actual number of applications in use, they provide a good understanding of the growing popularity of CEIs and the increasing demand for data interpretation and integration. However, even if people choose a theoretical framework, invest human and financial resources to collect data, and strive to implement the comprehensive evaluation process flawlessly, there might still be some defects in the evaluation process since there is no “gold standard” for determining factor weight in CEI. Despite such problems, CEIs are widely used by international organizations to measure economic, environmental, and social phenomena, thus serving as highly relevant tools [2].

Comprehensive evaluation or multi attribute decision making is still in a stage of blind exploration. This study aims to address the estimation of factor weights in comprehensive evaluation or MADM by incorporating statistics and informatics. Specifically, it focuses on how to assign scientific, objective, fair, and reasonable weight coefficients to the factors involved in constructing the comprehensive evaluation. The goal is to make a contribution to the methodology of constructing comprehensive evaluation indices. The authors believe that the relationship between the CEI and each evaluation factor, is similar to the relationship between work and energy in physics. The CEI represents the performance of each evaluation factor in comprehensive evaluation, and each evaluation factor is the source of abilities possessed by the CEI. There exists an underlying quantitative functional relationship between the CEI and the individual evaluation factors. In our current study, we establish a specific equation for this underlying function, but we have not yet found a method to solve it. So far, all mathematical approaches to exploring the CEI have not attempted to quantitatively solve this function. Instead, they have explored the correlation between the CEI and the individual evaluation factors from other perspectives. As a result, these non-quantitative methods that deviate from the underlying quantitative function have been in vain.

To address this issue, the author proposes a nonlinear fitting method based on the maximum information entropy [10,11] and the least squares residual, which is similar to weighted regression analysis. This method quantitatively fits the relationship between the CEI and the individual evaluation factors, treating the weight coefficient estimation and indicator fusion as a whole. This approach aims to solve the weights of the individual factors and fundamentally resolve the issues in the modeling process of the CEI. Section 2 specifies the definitions of the proposed maximum entropy-minimum residual model for comprehensive evaluation, and the related statistical tests in the new proposed model. Section 3 provides the definition of min-max normalization that is mainly used as a data preprocessing technique in this paper and gives an illustrative example to elaborate on the optimization results of the new model with DPS software V21.05. Section 4 addresses comparisons of the proposed model with some existing commonly used models in various cases.

## 2. Maximum Entropy-Minimum Residual Comprehensive Evaluation Index Model

### 2.1. Comprehensive Evaluation Index Model

Given *n* samples of *p* factors, let $x_{ij}$ denote the observation value of the *i*-th sample from the *j*-th factor, i=1,2,⋯,n,j=1,2,⋯,p. From the perspective of statistical models, constructing a multi-factor comprehensive evaluation index, or the so-called composite indicator, ${\left\{y_{i}\right\}}_{i=1}^{n}$, requires estimating the weight coefficients, $w_{j}$, of each factor in the following equation(1)$$y_{i}=x_{i1}w_{1}+x_{i2}w_{2}+\cdots +x_{ip}w_{p},$$
where $w_{1}+w_{2}+\cdots +w_{p}=1$, and $w_{j}\geq 0$. However, since there is no explicit dependent variable ($y_{i}$) for reference in the comprehensive evaluation model here, the empirical model proposed by this paper assumes that the composite indicator ($y_{i}$) is composed of *p* factors such that $y_{i}$ can be considered as the sum of theoretical indicator values from the *p* factors. Suppose that the *j*-th factor is assigned a weight $w_{j}$, the theoretical indicator value of each factor, also called by theoretical component denoted as $C_{ij}$, is given by $C_{ij}=y_{i}w_{j}$. The proposed model and the data fitting are based on $C_{ij}$, where $x_{ij}w_{j}$ is considered as the observed values of $C_{ij}$. Therefore, $y_{i}$ can also be expressed in the form of the sum of its theoretical components,(2)$$y_{i}=C_{i1}+C_{i2}+\cdots +C_{ip}.$$ Since the Equations (Equation 1) and (Equation 2) are equivalent from the left side, it is justified to assume that $x_{ij}w_{j}$ and $C_{ij}$ are in one-to-one correspondence. Remark that $x_{ij}w_{j}$ is not necessary to be equal to the true theoretical component $C_{ij}$. Otherwise, $x_{ij}$ will be exactly equal to $y_{i}$, which is not practical. Considering $x_{ij}w_{j}$ as the observed values of the theoretical components $C_{ij}$, we can empirically conclude that the closer the two values are, the more representative the CEI becomes.

We define the difference between the observed and the theoretical components as the residual, $\sigma_{ij}=x_{ij}w_{j}-C_{ij}$, which is considered as a fitted error of the *i*-th sample of the *j*-th factor. In a regression model, the parameter estimation is derived from minimizing the residual, or say the root mean square error of $y_{i}$, denoted by σ in our case,(3)$$\sigma =\sqrt{\frac{\sum_{i=1}^{n}\sum_{j=1}^{p}{\left(x_{ij}w_{j}-C_{ij}\right)}^{2}}{(n-1)p}}=\sqrt{\frac{\sum_{i=1}^{n}\sum_{j=1}^{p}{\left[\left(x_{ij}-y_{i}\right)w_{j}\right]}^{2}}{(n-1)p}}.$$ From Equation (Equation 3), the residual error $\sigma_{ij}$ can also be expressed as a weighted difference between the observed value $x_{ij}$ and the theoretical CEI $y_{i}$. The larger the weight coefficient of a factor, the greater its impact on the residuals, indicating that the factor is “more important”. This approach is essentially a weighted regression analysis process. It is consistent with the principle in multi-factor comprehensive analysis, where a larger weight coefficient for a factor means that the factor is more important. The fitted error term $\sigma_{ij}$ is assumed independent of other factors, since under ideal circumstances, the component of each individual factor can be considered independent of the other factors in order to measure a specific characteristic of the CEI. The root mean square error of Equation (Equation 3) is similar to the decomposition of the sum of squared errors in a regression model, and the corresponding degree of freedom in our case is (n−1)p. Equation (Equation 3) can be considered as an error function to be minimized when deriving the solution of weights. However, if the weight coefficient of the *j*-th factor in Equation (Equation 3) is equal to one and the weight coefficients of the rest of the factors are zeros, then $y_{i}$ is equal to the observed value $x_{ij}$ while the residual $y_{i}-x_{ij}$ is minimized to zero. Obviously, directly minimizing Equation (Equation 3) ends up with a trivial solution in which a certain weight is 1 and zero otherwise, making the corresponding factor as the CEI. Although, σ=0, in this special case, the resulting CEI is pointless.

### 2.2. Maximum Entropy-Minimum Residual (MEMR) Model for Comprehensive Evaluation

Entropy is a key concept in information theory, as a unit of measurement when quantifying the total amount of information in a dataset. The maximum entropy (ME) principle in [10] provides a method for reasoning from incomplete information. For problems in science, the more information we have, the better the models and theories we can construct. However, for a complex system, such as a multi-factor CEI model, developers never have enough information to unambiguously predict the probability distribution of the weights of individual factors. Indeed, constructing a CEI for such a complex system faces an underdetermined reasoning problem. When the ME principle is adopted by specifying available information as constraints, a model with the least bias can be chosen from all possible models consistent with the information. In establishing a CEI model, since the dimensions of the various indicators are inconsistent, the common approach is to preprocess the data using min-max normalization introduced in Section 3 or other normalization methods so that the normalized values of each factor fall between 0 and 1. We can treat these as ratio values between 0 and 1, representing the probability of a factor’s distribution within the evaluation range. ME inference is considered the correct way to estimate an unknown probability distribution given limited information (in the form of moments of the distribution) when entropy defined by [12] is chosen to effectively measure information for the probability distribution of random events [13]. Distributions with higher entropy implicate greater diversity and are more likely to be observed [13]. Thus, ME inference leads to the best probability estimate from the given information, with no need to assume any further knowledge beyond that in a set of constraints. According to ME inference, any other form will be based on nonoptimal inference, either using less information than that available or expressing unjustified bias.

The ME principle is thus adopted in the present study to determine weight coefficients of multiple factors in the construction of a CEI. If each weight coefficient follows a probability distribution for the importance of each factor, the ME form of the probability distribution can be achieved through maximization of Shannon’s information entropy, which can be expressed in the following formula containing weight coefficients $w_{j}$ of all concerned factors,(4)$$E=-\sum_{j=1}^{p}w_{j}\log\left(w_{j}\right),$$
in which information entropy is at the maximum when all weight coefficients are equal. When the weight coefficient is equal to 1 for one factor and 0 for each of the other factors, information entropy is at the minimum. This seems to contradict the optimization of a CEI by maximizing information entropy and minimizing residual error according to the ME principle.

The idea of using information entropy for CEI modeling is intuitive, similar to compression regression methods such as LASSO regression, where the optimization function for the CEI model is established in the following form,minf(σ,E)=min(σ−λE),
in which taking different values for λ leads to different optimization results. An appropriate λ may provide a model as required. However, the biggest problem with this approach is how to choose λ. At the same time, the objective function to be optimized is a hybrid because the dimensions of the residuals and information entropy are different. Combining them in the form of a sum or difference as the objective function is inappropriate, since the CEI model essentially aims to optimize the residuals defined by Equation (Equation 3). Another possible way to combine residual and entropy is through multiplication or division asminf(σ,E)=min(σ/E),
that simply scales the residuals up or down by several times, which is a linear relationship. The result is the same as optimizing based on the residual σ defined by Equation (Equation 3), and it does not yield the desired statistical model. Given that the previous two forms are not suitable for constructing the CEI model, a natural idea is to adopt an exponential function to establish the error function, using the residual σ as the base and information entropy *E* as the exponent, defined as(5)$$minf\left(\sigma ,E\right)=min\left(\sigma^{E}\right).$$ Since the dimensions of all concerned factors are not consistent, data must be properly normalized in advance, such as using the min-max normalization method, in order to construct a CEI or MADM model. The normalized dimension value of each factor is essentially a ratio between 0 and 1, and the resulting residual σ must be less than 1 accordingly, such that the error function value in Equation (Equation 5) becomes smaller when the entropy is larger. Therefore, the property of the error function in Equation (Equation 5) can be interpreted as the process of minimizing the residual σ within the space of information entropy *E*. For original data without normalization, each observed value can be divided by the maximum (absolute value) of all observed values to convert it to a ratio between 0 and 1 before the optimization begins. In summary, the maximum entropy-minimum residual (MEMR) comprehensive evaluation index model can be expressed as(6)$$min_{{\left\{w_{j}\right\}}_{j=1}^{p}}\left(\sigma^{E}\right),\;s.t.\;\sum_{j=1}^{p}w_{j}=1,\;w_{j}\geq 0,$$
where $w_{j},\;j=1,\;2,\;\cdots ,\;p$ with constraint is the optimization variable in this nonlinear regression equation model.

The nonlinear regression Equation (Equation 6) with constraints can be solved using a variety of nonlinear optimization methods with constraints. Since convergence speed varies with different optimization algorithms, it may take a long time to achieve the bootstrap sampling estimation of model weight parameters. Thus, it is worthy of searching for an algorithm with stable operation and fast speed. We try to use the sequential unconstrained minimization technique (SUMT) for optimization and obtain optimal results (shown in Section 3). The SUMT optimization algorithm has been included in DPS (data processing system) software package V21.05 [14]. In this paper, a CEI constructed with the proposed MEMR model is called MEMR index, and the corresponding weight coefficient is called MEMR weight. MEMR modeling actually uses least squares techniques to extract the common trends of various factors in the process of building a CEI model or conducting MADM while taking into account all factors (information) as comprehensively as possible. This is highly consistent with the motivation behind conducting comprehensive evaluation or MADM.

### 2.3. Statistical Tests in Comprehensive Evaluation Index Models

In constructing a CEI, the simplest approach is the equal weighting (EW) model, where all variables are assigned the same weight. If there are *p* variables, the weight coefficients for all factors are equal to $\frac{1}{p}$. Clearly, in actual comprehensive evaluation and MADM processes, the importance of each evaluation factor is not the same. Here, we use the equal weighting model as a reference (null model) for the CEI model, treating it as a submodel of the MEMR model, and performing statistical tests using a nested model approach. This means we can use statistical testing methods to compare the MEMR model with the EW model, i.e., to test whether the MEMR model can be simplified into the EW model (null hypothesis). The statistical test can be conducted using the *F*-test. The sum of squared residuals for each model is shown in Table 1, in which(7)$$\begin{array}{cc} RSS(w_{EW}) & =\sum_{i=1}^{n}\sum_{j=1}^{p}{\left(\frac{x_{ij}-y_{i}}{p}\right)}^{2},\;y_{i}=\sum_{j=1}^{p}\frac{x_{ij}}{p}, \end{array}$$(8)$$\begin{array}{cc} RSS(w_{MEMR}) & =\sum_{i=1}^{n}\sum_{j=1}^{p}{\left[\left(x_{ij}-y_{i}\right)\cdot w_{j}\right]}^{2},\;y_{i}=\sum_{j=1}^{p}x_{ij}w_{j}. \end{array}$$

Currently, statistical software such as Stata V16.0 and Prism V5.0 have applied analysis of variance (ANOVA) for statistical tests in nested nonlinear models; see [15]. In this paper, based on [16], the *F*-test statistic is expressed in terms of the sum of squared residuals and serves as a measure of the distance between the EW model and the MEMR model. A larger *F*-statistic indicates that the two models differ significantly, suggesting that the EW model does not describe the data well due to its equal weight coefficients for all factors. Therefore, the more general model, MEMR, is more suitable for comprehensive evaluation and MADM. The *F*-test statistic is a ratio of the last two mean square terms in Table 1, where the numerator is the difference in the variance terms between the two models, divided by the difference in the number of parameters between the models. The denominator is the mean square error of the MEMR model (the full model). The statistical test formula is as follows:(9)$$F=\frac{\frac{RSS\left({\hat{w}}_{EW}\right)-RSS\left({\hat{w}}_{MEMR}\right)}{p-1}}{\frac{RSS({\hat{w}}_{MEMR})}{(n-1)p}}.$$ The *F*-test statistic is defined the same way as in linear models. However, unlike in linear models, the *F*-distribution only holds approximately (it becomes closer as the sample size increases). Based on the *F*-value and its degrees of freedom, it is not difficult to obtain the significance probability, or *p*-value. From the *F*-statistic test, we can conclude that if the statistical test is not significant (for example, using a 0.05 significance level), we can accept the EW model as the CEI model. However, generally speaking, the MEMR model can always improve the effectiveness of comprehensive evaluation or MADM to some extent. If p<0.05, then the EW model may not be suitable as the CEI model, and the MEMR model should be adopted. This is because, in the CEI model, at least one of the weight coefficients is different from the others, making the optimization process meaningful. The weight coefficients generated through optimization significantly improve the closeness of the composite indicator to the observed values compared to the equal weighting coefficients.

In addition, we define the model optimization index *R* as the percentage decrease of the difference in the sum of squared residuals between the EW model and the MEMR model, compared to the sum of squared residuals of the EW model,(10)$$R=\frac{RSS\left({\hat{w}}_{EW}\right)-RSS\left({\hat{w}}_{MEMR}\right)}{RSS({\hat{w}}_{EW})}\times 100\%.$$ The meaning of *R* is that, after optimization, it measures the extent to which the CEI model has been improved compared to the null hypothesis, where each factor is assigned an equal weight. This is reflected by the percentage decrease in the sum of squared residuals, or it can be interpreted as the degree to which the goodness of fit has been improved.

Since the MEMR model is a nonlinear model, it is difficult to accurately estimate the standard errors of the weight coefficients. Here, we apply the Bootstrap method for estimation. For the multi-factor MEMR CEI model, a set of weight coefficients can be calculated for each Bootstrap sample. If we draw $n_{B}=1000$ Bootstrap samples, we can obtain 1000 sets of weight coefficients.

1.95% Confidence Interval of Weight Coefficients

Based on these 1000 sets of weight coefficients, the 2.5th percentile and the 97.5lth percentile are calculated to obtain the 95% confidence intervals for each weight coefficient. In case one focuses on the analysis of the impact of the weight coefficients, for a given weight coefficient, if the interval does not include $\frac{1}{p}$, then it can be considered that the influence of this factor is significantly too large or too small. In a comprehensive evaluation, a subset of the weights can be $\frac{1}{p}$, but this does not mean that we have completely equal weights. In this case, it is not appropriate to use equal weight to calculate the composite indicator in a multi-factor comprehensive evaluation.

2.95% Standard Error of Weight Coefficients and Coefficient of Variation

Based on certain samples drawn using Bootstrap sampling, each having the same sample size as the initial one, and for each sample, the mean and variance of the sample statistics are calculated. For instance, the sample means are denoted by ${\bar{x}}_{1},\;{\bar{x}}_{2},\;\cdots ,\;{\bar{x}}_{n_{B}}$, and the mean estimation is given by $\hat{\mu }=\sum_{i=1}^{n_{B}}\frac{{\bar{x}}_{i}}{n_{B}}$. According to the formula, $\sigma^{2}=\sum_{i}^{n_{B}}\frac{{\left({\bar{x}}_{i}-\hat{\mu }\right)}^{2}}{n_{B}-1}$, the coefficient of variation for each factor is given by $CV=\hat{\sigma }/\hat{\mu }$.

The DPS V21.05 system calculates the average coefficient of variation for each factor to examine the fluctuation range of the weight coefficients of each factor in the CEI model, as well as the stability of the weight coefficients during the comprehensive evaluation process. Based on the size of the coefficient of variation, models with smaller coefficients of variation are selected from several models as application references.

## 3. An Example for Optimization of MEMR Model

### 3.1. Min-Max Normalization

When constructing CEI, many evaluation factors are measured with different units. Sometimes, the factors are considered to be ‘good’, if a higher value indicates better performance. Otherwise, the factors are considered ‘bad’, such as ‘long-term unemployment’ and ‘obstruction of daily activities owing to chronic illness,’ for which higher values are associated with worse performance. To put all factors on a common dimension, data must be normalized using a rescaling technique. According to a factor that is considered ‘good’ or ‘bad’, the following normalization technique is called min-max normalization, which is popularly used for preprocessing data in comprehensive evaluation,(11)$$x_{ij}^{\prime }=\left\{\begin{array}{cc} \frac{x_{ij}-x_{min\;j}}{x_{max\;j}-x_{min\;j}} & x_{ij}\;\mathrm{is}\;a\;‘\mathrm{good}’\;\mathrm{factor},\\ \frac{x_{max\;j}-x_{ij}}{x_{max\;j}-x_{min\;j}} & x_{ij}\;\mathrm{is}\;a\;‘\mathrm{bad}’\;\mathrm{factor}. \end{array}\right.$$ All normalized factors are represented by ratios between 0 and 1, which correspond to the ‘worst’ and the ‘best’ performance, respectively. The case studies in this paper adopt the min-max normalization technique for simplicity. The DPS software V21.05 is capable of adopting other normalization techniques.

### 3.2. An Illustrative Example

For example, the min-max normalization is adopted when conducting a comprehensive evaluation analysis of the average per capita living consumption expenditure level of rural households in different regions of China in 2007 with the original data listed in the following Table 2, downloaded from https://download.csdn.net/download/weixin_42119432/12617871 (accessed on 8 January 2025).

After data normalization using Equation (Equation 11), the MEMR model is optimized with a sequential unconstrained minimization technique (SUMT) conducted through DPS software V21.05. The system first outputs the information entropy of the nonlinear MEMR model with constraints with the value of 2.9806, while the logarithm of the objective function is equal to −13.2752 and the residual error σ=0.0116, as listed in Table 3. The second part of the results is about the ANOVA table, where the null hypothesis $H_{0}$ is that the weight coefficient of the *j*-th factor is $\frac{1}{p}$, and the alternative hypothesis is that at least one factor *j* has a weight coefficient unequal to $\frac{1}{p}$. This part of results is interpreted from a statistical perspective in order to test whether there exists a difference among the weight coefficients of factors. If there is no significant difference (do not reject $H_{0}$), then an EW model can be used to construct the composite indicator. If the *p*-value is less than 0.05 (reject $H_{0}$), then the composite indicator needs to be calculated by the MEMR weights. In Table 3, *p*-value is 0.0002, implying that the CEI in this case should be calculated by the estimated weights of the factors instead of equal weights.

The third part of the output represents the weight coefficients of various factors, which serve as the core and focus of the comprehensive evaluation analysis model. These coefficients reflect both the contribution of each factor to the formation of the CEI and the importance of each factor in the CEI. They are the result of the interaction between individual factors and the CEI, derived under the condition of maximum entropy, minimizing fitting errors while accounting for central tendency, dispersion, and the interrelationships among multiple factors. The second column of the third part of Table 3 is the estimated weight coefficients, where a greater weight indicates that the corresponding factor is more important in comprehensive evaluation. In this case, based on weights, the two factors ‘Transportation and Communication’ and ‘Household Appliance’ have the greatest influence on the CEI. The second important factors are ‘Others’, ‘Housing’ and ‘Food’, while the factors such as ‘Entertainment’ and ‘Healthcare’ have a mild influence. The sample mean and standard deviation columns are estimated values calculated based on the results of Bootstrap sampling (1000 iterations in this case). The standard deviation here represents the standard error of the weight coefficients under the normality assumption. Using the standard error, the 95% confidence interval can be estimated. The “Median” and “95% Confidence Interval” are estimated based on the percentiles of Bootstrap sampling. Generally, when the 95% confidence interval exceeds $\frac{1}{p}$, the weight coefficient is considered significantly larger; otherwise, the weight coefficient is considered significantly smaller. For example, in this case, the confidence intervals for the weights of the ’Household Appliance’ and ’Transportation and Communication’ factors are [0.1263, 0.1884] and [0.1350, 0.1826], respectively, both greater than $\frac{1}{p}=0.125$. Therefore, these two factors can be considered significantly larger in a statistical sense.

Bootstrap sampling estimation serves two purposes: statistical testing and assessing the sensitivity of each weight coefficient. The smaller the range of the 95% confidence interval based on Bootstrap percentiles, the more robust the weight coefficient of the factor is in evaluating the composite indicator or making multi-attribute decisions. Conversely, a wider confidence interval suggests that the weight coefficient should be treated with more caution when estimating the composite indicator and making decisions. Using Bootstrap simulation sampling, we estimate the standard error of each factor’s weight coefficient. To examine the estimation accuracy, we can calculate the mean standard error for all factors, as well as the mean weight coefficient for each factor (equal to $\frac{1}{p}$). Subsequently, we can calculate the coefficients of variation (CV) for the weight coefficients. The CV is typically used to compare the dispersion of different datasets. A larger CV indicates greater variability and higher dispersion, while a smaller CV suggests less variability and lower dispersion. By comparing the CVs of different datasets, we can evaluate their stability and consistency.

Finally, based on the CEI values generated by the maximum entropy-minimum residual model for each sample, we can rank the evaluation indices from highest to lowest to obtain the ranking results for all samples. In this example, the estimated CEIs for each region, ranked from highest to lowest, are listed in the last part of Table 3. Based on the ranking results, a professional interpretation can be made along with a comparison of the positions of the evaluation objects. For example, in this case, Shanghai, Beijing, and Zhejiang rank as the top three, indicating the highest levels of consumption. They are followed by Jiangsu, Fujian, Guangdong, and Shandong, which have slightly lower consumption levels.

## 4. Comparisons of MEMR Model with Commonly Used Models

### 4.1. Technical Achievement Index

The original data for the construction of technical achievement index (TAI) by OECD [2] also adopted the min-max normalization, listed in Table A1 of the Appendix A for simplicity. Based on this data set, weight coefficients of eight TAI-based factors computed with the proposed MEMR model are compared with those computed with the methods of equal weighting (EW), principal component analysis (PCA) [17], factor analysis (FA) [18], and standard deviation (SD) [19], and criteria importance through intercriteria correlation (CRITIC) [20]. Through the correlation analysis as shown in Figure 1, only the MEMR weights and the first principal component are highly positively correlated.

PCA is an industry-accepted method to reduce data dimensionality. The example data results show that the trend in the magnitude of the MEMR weights is consistent with the factor loadings of the first principal component in PCA, with a correlation coefficient of 0.8263. This indicates that the MEMR weights are essentially a form of data dimensionality reduction method. The MEMR and PCA weights are mostly negatively correlated with the weight coefficients obtained from other weighting methods. This suggests that, although these weights explain the variation of the evaluation factors, they do not serve the purpose of dimensionality reduction or combining the individual factors into a composite indicator.

Here, we also plot the relationships between the composite indicator, factor weights, and normalized values obtained using the MEMR method, the entropy weighting (ET) method [19], the CRITIC method, and the SD method in the form of a line-bubble combination chart as Figure 2. In the chart, the sample identification label (Region) is plotted on the horizontal axis, while the vertical axis represents the CEI values and the normalized values of individual factors, along with the factor weights displayed as bubbles. The diameter of the bubbles is uniformly defined as $0.025\times \frac{w_{j}}{w_{max}}$, where $w_{j}$ is the *j*-th factor weight coefficient, and $w_{max}$ is the maximum weight coefficient. The chart conveys four main points:Ranking and similarity of evaluation objects: the closer the points are, the more similar the evaluation objects are;Trend of the CEI line: the variation trend of the CEI along the line;Distribution of normalized values of each factor: the positioning of the normalized values of each factor around the CEI line indicates that the more concentrated the values, the better the ranking or evaluation effect;Factor weight coefficients: each factor is represented by a different color, with the size of each bubble indicating the magnitude of the weight coefficient. The larger the weight coefficient, the larger the bubble, and vice versa. This helps identify the distribution trends of different factors and their consistency with the CEI line.

As shown in Figure 2, in the model established using the MEMR method, factors with larger weights are distributed on both sides of the CEI line, while factors with smaller weight coefficients are located farther from the CEI line. The distribution of factor weight magnitudes aligns with the trend of the CEI values. This clearly reflects the representativeness of the CEI for the major factors with larger weight coefficients. However, the ET method, CRITIC method, and SD method do not exhibit a similar trend. This suggests that the methods of estimating factor weights for comprehensive evaluation using entropy weighting, criteria importance through intercriteria correlation, and standard deviation require further consideration. Remark that in the “Handbook on Constructing Composite Indicators” compiled by the OECD/EU-JRC [2], ET and CRITIC weighting methods are not mentioned.

### 4.2. Virtual Example Presented by an Orthogonal Table

An orthogonal table provides a virtual dataset comprising 24 orthogonal samples of four factors, each having 2, 3, 4, and 8 levels, respectively, listed in Table A2 of the Appendix A, generated by the DPS software V21.05. All factors have no mutual correlation and hence are independent of one another. In the meanwhile, the importance of the qualitative factors can be intuitively distinguished by the number of levels since more levels imply more information. The original data are normalized with the min-max method. Based on this virtual data set after normalization, the weight estimates of each factor computed by MEMR, ET, SD, CRITIC, and PCA are listed in Table 4.

The PCA weighting method is used to correct for overlapping information between two or more correlated factors, rather than to measure the theoretical importance of the related indicators. If there is no correlation between the factors, this method should not be used to estimate weights, and in this case, PCA is not suitable for estimation. As the analysis results show, the normalized eigenvector used as the weight coefficient is a 4lth-order identity matrix, with all diagonal elements equal to 1 and all other elements equal to 0. The weight coefficients obtained using the ET method exhibit a trend where the weight coefficient decreases as the number of levels of a factor increases. This contradicts our intuitive analysis of the amount of information contained in the data. Specifically, if we treat each level of a factor as an ordinal variable, variables with more levels should typically have greater information content, providing more potential for analysis and inference. For example, $x_{4}$ has 8 levels, which is close to a quantitative indicator, and it should theoretically contain more information, thus having a larger weight in the comprehensive evaluation index. However, the weight coefficient here is less than half that of the two-level factor $x_{1}$. Moreover, since the correlation coefficients between the individual factors are zero, the weight results calculated using the CRITIC method are the same as those obtained using the SD method. Their results show a similar trend as the ET method, where the weight coefficient decreases as the number of levels of a factor increases. Therefore, this suggests that the ET, SD, and CRITIC methods, as ways of calculating weight coefficients, need further discussion.

Using the MEMR weights proposed in this paper, the trend shows that the weight coefficient increases as the number of levels of a factor increases. This is consistent with our intuitive understanding that variables with more levels typically contain more information than those with fewer levels. Additionally, with this orthogonal table as an example, if each factor in the orthogonal table adopts a *z*-score normalization by subtracting the mean and dividing by the standard deviation, then the mean of each factor will be zero, and the standard deviation will be 1. In this case, the weight coefficients of each factor, calculated according to MEMR model Equation (Equation 6), will be the same. This further illustrates that orthogonal tables have the characteristics of being uniformly distributed, orderly, and comparable.

### 4.3. Example for Comprehensive Development Status of Enterprises

In order to improve the investment efficiency, a large investment firm intended to evaluate 15 companies with 10 indicators that reflect the enterprise development status [21]. The evaluation results after the min-max normalization are included in Table 5.

Ref. [21] used various BoD methods and their improvements to construct weights and calculate the CEI. The corresponding BoD methods and the results of other methods are shown in Table 6.

The non-compensatory BoD method introduced by [6] suffers from a significant 0–1 weight issue and often requires compensation rules of zero weights in the process of weighting. The BoD models without or with weight compensation are referred to as non-compensatory or compensatory BoD. The CEI results of the non-compensatory BoD method for evaluating the comprehensive development status of enterprises are listed in Table 6. From the results, they cannot be used for ranking the enterprises. Ref. [6] gave five weight compensation methods by adding a rigid score compensation constraint to the non-compensatory BoD to address over-flexibility issues. Setting the constraint value to 0.85 as an example of absolute compensation, it is impossible to rank all the concerned companies (Table 6).

The generalized BoD method [22] relaxes all computational symbols to the general summation operator “⊕” and the general multiplication operator “⊗” by means of a mathematical programming approach with geometric weighting aggregation patterns but is still unable to rank the companies meaningfully. Ref. [23] proposed a concept of “good indices” and “bad indices” from the positive and negative aspects of the two-set BoD weighting method. This approach adds more information to a more discriminative bidirectional composite indicator in the BoD analysis. After the “good” and “bad” indices computed with the two-set BoD method are adjusted with the parameter λ=0.5, the resultant composite indicators remain undesirable for ranking the companies (Table 6).

If the importance levels of evaluation factors are known, the method of multi-criteria ABC analysis [24], close to the BoD analysis, can be adopted. Assuming the decreasing importance of each factor for the development status of those companies from left to right, the ABC analysis results in the rankings of the 15 companies from 0.55 to 1 (Table 6). The composite indicators computed with the equal weighting method and the proposed MEMR method range from 0.47 to 0.62 and 0.46 to 0.63 (Table 6), respectively.

From the table, none of the BoD methods are suitable for the comprehensive evaluation of this dataset. A correlation analysis of the CEI computed with the ABC, EW, and MEMR methods shows that the correlation coefficients between the ABC method and the EW method, as well as between the ABC method and the MEMR method, are −0.1694 and −0.2934, respectively, both indicating negative correlations. The correlation coefficient between the MEMR method and the EW method is 0.9801, indicating a high positive correlation. This suggests that the ABC method does not reflect the trend of the composite indicator for each enterprise, whereas the composite indicator obtained through MEMR reflects the overall trend of each enterprise and can better rank the 15 enterprises.

## 5. Conclusions

Multi-factor comprehensive evaluation is a widely used measurement method, whose results serve as an important auxiliary tool for decision-making. A key problem concerns how to assign weights to each factor when constructing a comprehensive evaluation index. The fair and reasonable calculation of factor weights can provide a reliable foundation for decision-making. An unreasonable indicator, nevertheless, can make it difficult to provide objective evaluation results for decision-making bodies and even result in misleading issues or controversy.

This study proposes an MEMR model based on the maximum entropy–minimum residual principle, directly applying the relationship estimation between multi-factor weights and the composite indicator. Based on the least-squares fitting of the theoretical component and the observed component, the model value of the theoretical component is the closest to the observed value of the actual component. The resulting CEI represents the central tendency and dispersion degree of each factor to the greatest extent, making it the most representative. The optimization process that minimizes the residuals while maximizing the information entropy is a method of extracting the common features of multiple factors or attributes under the premise of retaining as much information as possible. In a sense, this is the process of concentrating on a certain trend based on adopting as many suggestions as possible and synthesizing the common points expressed by each factor, that is, seeking unity while preserving diversity. This aligns with the basic principles of comprehensive evaluation and multiple attribute decision-making.

Our proposed model can reflect the influence of each factor indicator better than the factor weight in the widely used equal weighting method and is closer to the actual meaning expressed by CEI. In the worst case, our model has the same evaluation outcome as the equal weighting method. PCA mainly considers the correlations between each factor. When the correlation coefficient between factors is small, the results of PCA are not very accurate. In addition, there is no standard regarding how many principal components need to be selected. When the number of principal components is different, the comprehensive scores might also be different. In this case, the evaluation object and its stakeholders will question the “rationality” of weight assignment.

The entropy weighting method, standard deviation method, and CRITIC method mainly estimate weight based on the dispersion degree of each factor, without considering the influence of the central tendency of each factor. These weighting methods do not take into account the quantitative relationship between each factor, its weight, and the estimated composite indicator. Instead, they focus more on presenting the data from a certain external characteristic to explain a particular phenomenon. They display data according to certain needs instead of deeply revealing the information contained within the data. There is no standard for when and how to choose these weighting methods, and they are not universally suitable for all comprehensive evaluation processes of quantitative factors. The problem becomes particularly salient when applying BoD for comprehensive evaluation.

Our proposed maximum entropy-minimum residual composite evaluation analysis modeling technique generates a composite evaluation index through numerical optimization, based on considering as much multi-factor information as possible while refining the central tendency of the factors. In the field of multivariate statistical analysis, the MEMR model employs nonlinear optimization techniques, allowing high-dimensional data to be more effectively mapped to one-dimensional space than principal component analysis. In various scenarios including big data analysis, we can apply the MEMR model for dimensionality reduction analysis of multivariate data.

The MEMR model is a new approach that combines information science and statistics. Accordingly, further theoretical research is needed in the following aspects.

Inference and estimation of MEMR model parameters: this includes improving the current nonlinear numerical optimization method and inventing new methods in order to improve the effectiveness and robustness of the optimization process; calculating inference accuracy and the confidence interval of parameter estimation; and further determining how to test the model hypothesis, diagnosing goodness of fit, and determining the rationality of the residual error;Estimation of weight coefficients in a MEMR model with attribute variables;Application to weight estimation in multiple attribute decision making (MADM): in MADM, it is necessary to consider how to evaluate the accuracy of each factor weight estimated by the MEMR model for predicting future data or making new observations;Extensions of the MEMR model: the model can be generalized to the problem involving panel data or time series data; we also need to consider the applications of the MEMR model in different fields, such as medicine, ecology, and finance, and how to customize and improve the model according to the needs of specific fields; how to adjust the model if considering a scenario where the expert defines factor weights in advance.

## Figures and Tables

**Figure 1 entropy-27-00203-f001:**
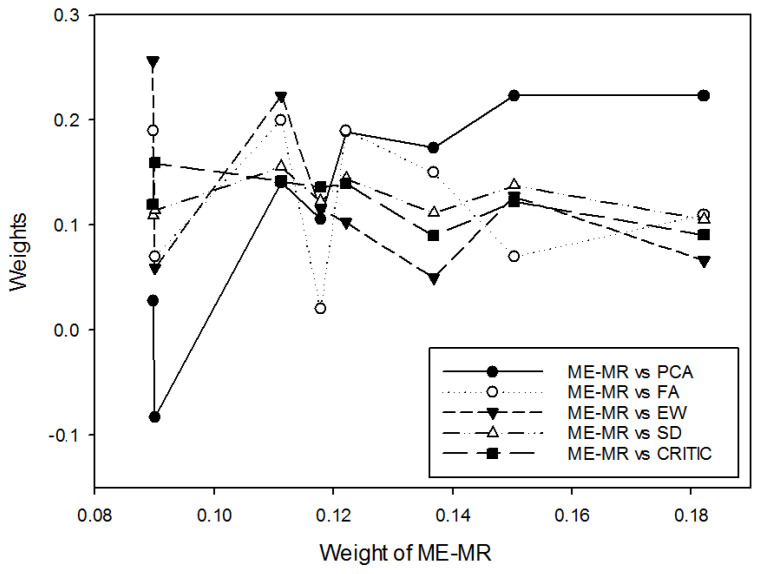
Weight coefficients of various weighting methods for the technological achievement index data.

**Figure 2 entropy-27-00203-f002:**
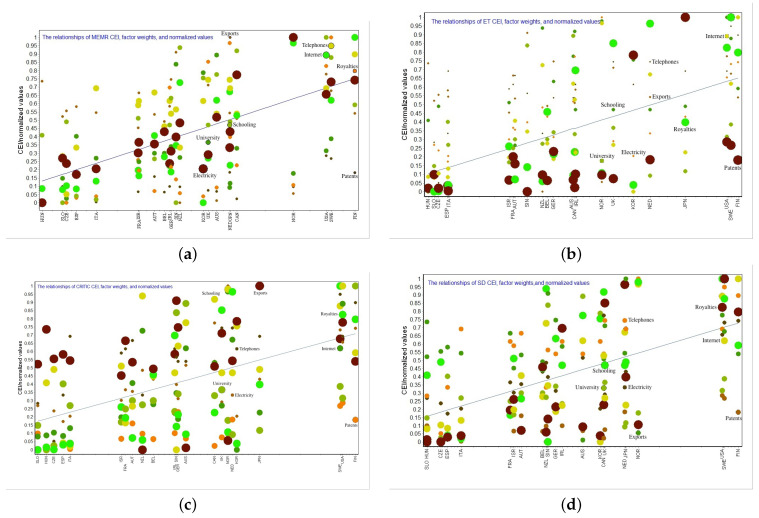
Relationship diagram of composite indicator estimates and factor weights and normalized values for different methods: (**a**) The relationships of MEMR CEI, factor weights, and normalized values. (**b**) The relationships of ET CEI, factor weights, and normalized values. (**c**) The relationships of CRITIC CEI, factor weights, and normalized values. (**d**) The relationships of SD CEI, factor weights, and normalized values.

**Table 1 entropy-27-00203-t001:** Analysis of variance in multi-factor comprehensive evaluation index models.

Source of Variation	Sum of Squared Residuals	Degree of Freedom	Mean Square
EW Model	$RSS(w_{EW})$	np−1	$\frac{RSS(w_{EW})}{np-1}$
MEMR Model	$RSS(w_{MEMR})$	(n−1)p	$\frac{RSS(w_{MEMR})}{(n-1)p}$
EW − MEMR	$RSS\left(w_{EW}\right)-RSS\left(w_{MEMR}\right)$	p−1	$\frac{RSS\left(w_{EW}\right)-RSS\left(w_{MEMR}\right)}{p-1}$

**Table 2 entropy-27-00203-t002:** Average per capita living consumption expenditure of rural households in different regions of China (2007).

Region	Food	Clothing	Housing	HA ^1^	TC ^2^	EM ^3^	HC ^4^	Others
Beijing	2132.51	513.44	1023.21	340.15	778.52	870.12	629.56	111.75
Tianjin	1367.75	286.33	674.81	126.74	400.11	312.07	306.19	64.3
Hebei	1025.72	185.68	627.98	140.45	318.19	243.3	188.06	57.4
Shanxi	1033.68	260.88	392.78	120.86	268.75	370.97	170.85	63.81
Inner Mongolia	1280.05	228.4	473.98	117.64	375.58	423.75	281.46	75.29
Liaoning	1334.18	281.19	513.11	142.07	361.77	362.78	265.01	108.05
Jilin	1240.93	227.96	399.11	120.95	337.46	339.77	311.37	87.89
Heilongjiang	1077.34	254.01	691.02	104.99	335.28	312.32	272.49	69.98
Shanghai	3259.48	475.51	2097.21	451.4	883.71	857.47	571.06	249.04
Jiangsu	1968.88	251.29	752.73	228.51	543.97	642.52	263.85	134.41
Zhejiang	2430.6	405.32	1498.5	338.8	782.98	750.69	452.44	142.26
Anhui	1192.57	166.31	479.46	144.23	258.29	283.17	177.04	52.98
Fujian	1870.32	235.61	660.55	184.21	465.4	356.26	174.12	107
Jiangxi	1492.02	147.71	474.49	121.54	277.15	252.78	167.71	61.08
Shandong	1369.2	224.18	682.13	195.99	422.36	424.89	230.84	71.98
Henan	1017.43	189.71	615.62	136.37	269.46	212.36	173.19	62.26
Hubei	1479.04	168.64	434.91	166.25	281.12	284.13	178.77	97.13
Hunan	1675.16	161.79	508.33	152.6	278.78	293.89	219.95	86.88
Guangdong	2087.58	162.33	763.01	163.85	443.24	254.94	199.31	128.06
Guangxi	1378.78	86.9	554.14	112.24	245.97	172.45	149.01	47.98
Hainan	1430.31	86.26	305.9	93.26	248.08	223.98	95.55	73.23
Chongqing	1376	136.34	263.73	138.34	208.69	195.97	168.57	39.06
Sichuan	1435.52	156.65	366.45	142.64	241.49	177.19	174.75	52.56
Guizhou	998.39	99.44	329.64	70.93	154.52	147.31	79.31	34.16
Yunnan	1226.69	112.52	586.07	107.15	216.67	181.73	167.92	38.43
Tibet	1079.83	245	418.83	133.26	156.57	65.39	50	68.74
Shaanxi	941.81	161.08	512.4	106.8	254.74	304.54	222.51	55.71
Gansu	944.14	112.2	295.23	91.4	186.17	208.9	149.82	29.36
Qinghai	1069.04	191.8	359.74	122.17	292.1	135.13	229.28	47.23
Ningxia	1019.35	184.26	450.55	109.27	265.76	192	239.4	68.17
Xinjiang	939.03	218.18	445.02	91.45	234.7	166.27	210.69	45.25

^1^ Household Appliance; ^2^ Transportation and Communication; ^3^ Entertainment; ^4^ Healthcare.

**Table 3 entropy-27-00203-t003:** The summary of MEMR modeling results through DPS software V21.05 with SUMT.

Data Normalization (Min-Max) and Transformation								
log (objective function)				−13.2752				
Residual (σ)				0.011633				
Information entropy (*E*)				2.980593				
**ANOVA table**								
**Sources of Variation**		**SS ^1^**	**df ^2^**	**Mean square**				
EM model		0.0366	247	0.0001				
MEMR model		0.0325	240	0.0001				
EM − MEMR		0.0041	7	0.0006				
*F*-value = 4.3389, *p*-value = 0.0002								
Optimization index = 11.2334%								
**Weight coefficients estimated by Bootstrap sampling (1000 times)**								
**Factors**	**Weights**	**Mean**	**Std r ^3^**	**Median**	**95% Confidence Interval**			
Food	0.1171	0.1158	0.0071	0.1155	0.0958	0.1392		
Clothing	0.1039	0.1048	0.0102	0.1049	0.0729	0.1369		
Housing	0.1186	0.1181	0.0088	0.1177	0.0879	0.1514		
HA ^4^	0.1580	0.1572	0.0101	0.1574	0.1263	0.1884		
TC ^5^	0.1612	0.1611	0.0082	0.1615	0.1350	0.1826		
EM ^6^	0.1096	0.1097	0.0088	0.1092	0.0793	0.1454		
HC ^7^	0.1072	0.1073	0.0075	0.1075	0.0842	0.1345		
Others	0.1245	0.1260	0.0146	0.1248	0.0843	0.1654		
Entropy	2.9806	2.9766	0.0062	2.9769	2.9465	2.9946		
**Sample rank and comprehensive evaluation index (CEI)**								
**Region**	**Rank**	**CEI**	**Region**	**Rank**	**CEI**	**Region**	**Rank**	**CEI**
Shanghai	1	0.9782	Hunan	12	0.2279	Sichuan	22	0.1493
Beijing	2	0.7264	Heilongjiang	13	0.2273	Qinghai	23	0.1429
Zhejiang	3	0.7146	Hubei	14	0.2179	Xinjiang	24	0.1225
Jiangsu	4	0.4529	Shanxi	15	0.1851	Guangxi	25	0.1219
Fujian	5	0.3313	Hebei	16	0.1828	Chongqing	26	0.1194
Guangdong	6	0.3205	Jiangxi	17	0.1699	Yunnan	27	0.1133
Shandong	7	0.3	Anhui	18	0.1661	Hainan	28	0.1124
Liaoning	8	0.2837	Henan	19	0.1659	Tibet	29	0.1044
Tianjin	9	0.2751	Ningxia	20	0.1547	Gansu	30	0.0621
Inner Mongolia	10	0.2513	Shaanxi	21	0.1509	Guizhou	31	0.0298
Jilin	11	0.2385						

^1^ Sum of Squares; ^2^ Degrees of Freedom; ^3^ Standard Deviation; ^4^ Household Appliance; ^5^ Transportation and Communication; ^6^ Entertainment; ^7^ Healthcare.

**Table 4 entropy-27-00203-t004:** Weight Coefficients of several weighting methods for virtual sample data in the orthogonal table.

Factor	Levels	MEMR	ET	SD	CRITIC	PCA
$x_{1}$	2	0.2040	0.3850	0.3109	0.3109	1
$x_{2}$	3	0.2469	0.2567	0.2538	0.2538	0
$x_{3}$	4	0.2642	0.2082	0.2317	0.2317	0
$x_{4}$	8	0.2849	0.1502	0.2035	0.2035	0

**Table 5 entropy-27-00203-t005:** Normalized data of the comprehensive development status of 15 enterprises.

Enterprise	$x_{1}$	$x_{2}$	$x_{3}$	$x_{4}$	$x_{5}$	$x_{6}$	$x_{7}$	$x_{8}$	$x_{9}$	$x_{10}$
1	1.0	0.1	0.2	0.3	0.4	0.5	0.6	0.7	0.8	0.9
2	0.9	0.2	0.1	0.2	0.3	0.4	0.5	0.6	0.7	0.8
3	0.8	0.9	0.7	0.1	0.2	0.3	0.4	0.5	0.6	0.7
4	0.7	0.8	0.9	0.3	0.1	0.2	0.3	0.4	0.5	0.6
5	0.6	0.7	0.8	0.9	1.0	0.1	0.2	0.3	0.4	0.5
6	0.5	0.6	0.7	0.8	0.9	1.0	0.1	0.2	0.3	0.4
7	0.4	0.5	0.6	0.7	0.8	0.9	1.0	0.1	0.2	0.3
8	0.3	0.4	0.5	0.6	0.7	0.8	0.9	1.0	0.1	0.2
9	0.2	0.3	0.4	0.5	0.6	0.7	0.8	0.9	0.6	0.1
10	0.1	0.2	0.3	0.4	0.5	0.6	0.7	0.8	0.9	1.0
11	0.8	0.3	0.2	0.5	0.6	0.5	0.8	0.6	0.4	0.7
12	0.9	1.0	0.2	0.5	0.6	0.5	0.8	0.6	0.4	0.7
13	0.8	0.2	1.0	0.6	0.6	0.5	0.8	0.6	0.4	0.7
14	0.8	0.2	0.6	1.0	0.6	0.5	0.8	0.6	0.4	0.7
15	0.8	0.2	0.6	0.2	0.6	0.5	0.8	0.6	1.0	0.7

**Table 6 entropy-27-00203-t006:** The comprehensive evaluation indices of 15 enterprises computed with various methods.

Enterprise	Non-Compensatory BoD	Compensatory BoD	Generalized BoD	Two-Set BoD	ABC	EW	MEMR
1	1.0	0.9548	1.0	0.5	1.00	0.55	0.5529
2	1.0	0.9724	1.0	0.5	0.95	0.47	0.4692
3	1.0	0.9985	1.0	1.0	0.90	0.52	0.5007
4	1.0	0.9985	1.0	0.5057	0.85	0.48	0.4604
5	1.0	1.0	1.0	0.5	0.80	0.55	0.5478
6	1.0	0.9985	1.0	0.5	0.75	0.55	0.5573
7	1.0	NULL ^1^	1.0	0.6264	0.70	0.55	0.5575
8	1.0	1.0	1.0	0.5	0.65	0.55	0.5634
9	1.0	0.9868	1.0	0.5	0.60	0.51	0.5223
10	1.0	NULL ^1^	1.0	0.5	0.55	0.55	0.5621
11	1.0	0.385	0.9938	0.5	0.80	0.54	0.5449
12	1.0	NULL ^1^	1.0	0.5	0.90	0.62	0.6111
13	1.0	1.0	1.0	0.5	0.80	0.62	0.6228
14	1.0	1.0	1.0	0.5	0.63	0.62	0.6268
15	1.0	1.0	1.0	0.5	0.80	0.60	0.6020

^1^ No solution.

## Data Availability

A part of the dataset utilized in this paper is openly available in CSDN Repository at https://download.csdn.net/download/weixin_42119432/12617871 (accessed on 1 January 2025). Other data are available within the article.

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
