# Peer review of "Maximum Entropy-Minimum Residual Model: An Optimum Solution to Comprehensive Evaluation and Multiple Attribute Decision Making"

_entropy, 2025, doi:10.3390/e27020203_

Round 1

Reviewer 1 Report

Comments and Suggestions for Authors

he paper contributes with insights and ideas into an important area due to its application areas and impact on planning decisions and large scale resource allocations.

As stated in the paper, CEI is to a large extent still in a stage of exploration, but at the same time extensively used in high impact decision making bodies, which should call for a greater interest in the area.

The paper itself is highly feasible to be published in the Entropy journal. It proposes a way of practically quantitatively relating a composite indicator with its evaluation attributes by means of maximising entropy and minimising the residuals.

* Line 49, please explain the "0-1 weight problem" here, are there particular circumstances in the underlying data giving rise to this?

* Line 246, "The weight 248 coefficients generated through optimization significantly improve the closeness of the composite indicator to the observed values", and then on line 117 the presentation of "observed values", please explain why x_ij*w_j is the observed value and not x_ij. To me it seems more intuitive that x_ij is the observed indicator value, which cause some confusion. 

* Line 266, and the 95% confidence interval. Is it sufficient for one weight coefficient that 1/p does not fall within the interval? I am thinking that a subset of weights can be 1/p but that does not entail that we have fully equal weights. 

* On a side note, when conforming to min-max normalization, do we face a situation where the CI is sensitive to what objects that are to be estimated through a CI. That is, if you remove or add "extreme" objects that will alter the min/max, then the resulting CI will be unintuitively affected? This is typically one property of min-max normalization but it might be of lesser concern here.

Reviewer 2 Report

Comments and Suggestions for Authors

The authors of the manuscript propose a method to find an optimal solution to the problem of assigning weights to attributes in a MADM problem, or to criteria in an MCDM problem.

The idea is original and well founded. However, the paper suffers from a lack of rigor in some aspects that should be correctly formulated to guarantee the validity of the results obtained.

From the outset it cannot be assumed that Cij = yi x wj. If Cij are the theoretical components, on what basis is this statement justified? Under what conditions would this be true?

The F-test assumes that the variables being compared follow a Normal distribution. It would be necessary to justify that the conditions are met so that the sum of squared residuals for each model are Normal.

The paragraph "Traditional parameter inference mainly relies on the Central Limit Theorem...follows a Normal distribution" in line 258 is incorrect and there is no need to incorporate it to justify what follows, which is correct. I suggest the authors delete that paragraph, which does not apply to any sampling distribution but only to that of sample means and sums.

It is unnecessary to cite references in the redundant way that authors do, indicating the name of the authors and the reference number. They should adopt the style indicated by the paper.

Lastly, there is a duplicate reference. References [6] and [20] are the same.

Round 2

Reviewer 2 Report

Comments and Suggestions for Authors

The authors have taken into account all my recommendations and have appropriately incorporated them into the manuscript. I think that in its current form the paper has enough scientific rigor to be published in Entropy.